# Automatic design of mechanical metamaterial actuators

Silvia Bonfanti[1], Roberto Guerra[1], Francesc Font-Clos[1], Daniel Rayneau-Kirkhope[1] &
Stefano Zapperi[1,2✉]

Mechanical metamaterial actuators achieve pre-determined input–output operations exploiting architectural features encoded within a single 3D printed element, thus removing the need for assembling different structural components. Despite the rapid progress in the field, there is still a need for efficient strategies to optimize metamaterial design for a variety of functions. We present a computational method for the automatic design of mechanical metamaterial actuators that combines a reinforced Monte Carlo method with discrete element simulations. 3D printing of selected mechanical metamaterial actuators shows that the machine-generated structures can reach high efficiency, exceeding human-designed structures. We also show that it is possible to design efficient actuators by training a deep neural network which is then able to predict the efficiency from the image of a structure and to identify its functional regions. The elementary actuators devised here can be combined to produce metamaterial machines of arbitrary complexity for countless engineering applications.

[1] Center for Complexity and Biosystems, Department of Physics, University of Milan, via Celoria 16, Milano 20133, Italy. [2] CNR - Consiglio Nazionale delle Ricerche, Istituto di Chimica della Materia Condensata e di Tecnologie per l'Energia, Via R. Cozzi 53, Milano 20125, Italy. ✉email: stefano.zapperi@unimi.it

**M**echanical metamaterials are a novel class of artificial materials engineered to have exceptional properties and responses that are difficult to find in conventional materials[1–5]. Their stiffness, strength-to-weight ratio[6–8], elastic response[9], Poisson's ratio[10–14], energy trapping[15,16], and fracture resistance[17,18] can be tuned to match or exceed those found in standard materials. The increasing interest in metamaterials is stimulated by the recent advances in digital manufacturing technologies e.g. 3D printing and automated assembly, which enable rapid manufacture of such material structures with the removal of many of the constraints in scale and geometry at lower and decreasing cost[19].

Metamaterials derive their properties not from the inherent nature of the bulk materials, but from their artificially designed internal geometry composed of multiple sub-elements, or cells, which are usually arranged in repeated regular patterns[20,21]. Recent papers have also demonstrated the possibility to introduce an increasing degree of disorder in the mechanical metamaterial, without losing effectiveness[22–25]. Since cells can be designed and placed in many different ways, the resulting structure can display multiple degrees of freedom, giving rise to a variety of unusual physical properties which then find natural applications in industrial design, as architectural motifs or reinforcement patterns for textiles, beams and other objects.

Interesting examples of shape changing structures are based on origami and kirigami which use folds and cuts to program shape changes[26–29] with several engineering applications such as in aerospace, for the satellite solar panel deployment[30]. Origami-based[31–33] and kirigami-based[34] structures provide inspiration for the design of mechanical metamaterials. In this context, a recent paper explored the folding behavior of prismatic building blocks with controllable multifunctionality and applied local actuation patterns to study their mechanical behavior[35].

Metamaterials can also be considered as true machines[36], able to accomplish mechanical functions through the transformation of input stimuli into a programmable set of outputs. In this case, constituent cells work together in a well-defined manner to obtain the final controlled directional macroscopic movement. Metamaterial machines can be exploited as mechanical actuators for human-machine interactions or as interactive/responsive components in robotics. Conventional design strategies for metamaterial structures and machines are often based on manual operations, which work reasonably well in specific conditions but are not guaranteed to yield maximum efficiency for all cases.

To overcome the limitations of manual design, we propose an automatic optimization method for the automatic design of mechanical metamaterial actuators (MMA), searching for the optimal output response to an applied input through iterative modifications of the structure. Generative methods based on optimization algorithms have been used to design allosteric materials[37–39], fiber-reinforced actuators[40] and to choose the optimal cell geometry in periodic metamaterial lattices[41,42]. Our method deals instead with ordered or disordered[22,25] metamaterials and selects the optimal output response to a given applied input through iterative modifications by removal or reinsertion of beams. We show that this optimization process can be efficiently realized coupling the optimization algorithm with discrete element simulations or with a suitably trained deep neural network.

## Results

**Automatic design under displacement input/output**. We consider two prototypical actuators, the first in which the desired input and output are orthogonal, $\mathbf{t}_{inp} = -\hat{y}$, $\mathbf{t}_{out} = -\hat{x}$ (see Fig. 1), and a second one in which they are anti-parallel, $\mathbf{t}_{inp} = -\hat{y}$, $\mathbf{t}_{out} = \hat{y}$. In Supplementary Fig. 1, we report the evolution of $\eta$ during the MC dynamics for several realizations of the orthogonal-functionalized MMAs (traces for the anti-parallel case are similar). An example of generation of such structure during the optimization steps can be visualized in Supplementary Movie 1. We note that, after the annealing phase, efficiency tends to evolve in steps, interspersed by noisy parts and by plateaus. Steps can occur when a mechanism that engages the desired response is finally triggered, while plateaus indicate a structure whose efficiency is robust against the removal or addition of one or more bonds. Hence, we have selected our most efficient reference samples from the configurations laying in these plateaus of $\eta$. We obtain a large variability of the final $\eta$ values, indicating a possible trapping in local minima that require thermally activated excitation to escape. Because of this, and depending on the post-annealing conditions, the exploration of the whole phase-space of the network can take very long times, which further increase with the network size.

The use of a simplified model within the minimization algorithm requires further validation through more refined simulations. We thus have converted our structures to a FEM mesh, in order to simulate the realistic response of the material (see "Methods"). Simulation results for orthogonal and anti-parallel motion are reported in Fig. 2 and Supplementary Fig. 3, respectively. Note that the achievable efficiency can easily approach and exceed the value obtained by the human-designed structures, reported on the left panels. In all the cases considered, the model-calculated efficiencies were validated by FEM simulations, with minimal corrections on $\eta$ with respect of DEM estimates. It is possible to obtain information about the stress propagation along the network from FEM or DEM simulations and thus identify the regions mostly involved in the mechanism actualization. In this respect, we note that the machine-generated structures are characterized by a broad distribution of internal stresses, indicating a collective engagement of the network. Conversely, in the human-designed structures the stress highlights the few pivot points employed to perform the movement.

**Size scaling of the algorithm**. We investigate the scaling of the algorithm from an empirical perspective. To do so, we prepare six different lattices of increasing size, starting at $N_b = 172$ bonds and up to $N_b = 885$ bonds. Following the displacement-based protocol, we fix an input displacement of 1% relative to the total side length of the lattice, and a desired output direction perpendicular to the input, as in the example in Fig. 1. We run our algorithm for a total of $10 \cdot N_b$ accepted MC steps, decreasing the temperature from 0.06 down to 0, followed by $2 \cdot N_b$ MC steps at zero temperature. We generate 100 different structures for each lattice size, totaling 600 structures. For each structure, we measure its total execution time $T$ and its efficiency $\eta$ (see Fig. 3).

The scaling exponent $\alpha$ is obtained by fitting a liner regression model between $\log T$ and $\log N_b$. We obtain an estimate of $\hat{\alpha} = 2.97$. The efficiency of the obtained configurations increases with $N_b$ for the smallest lattice sizes, to then reach a plateau at larger sizes. We notice that the Monte Carlo procedure was designed to scale with $N_b$ as well, so the relaxation part of the algorithm would account for the remaining $N_b^2$ scaling. Although some computational details or the usage of faster CPU equipment could lower the reported execution times, there is an upper bound on the lattice sizes that can be investigated with the present algorithm. Still, our implementation allows us to find efficient structures in less than roughly one day for lattice sizes of around 700 bonds, which is within the range of interest for mechanical metamaterial actuators.

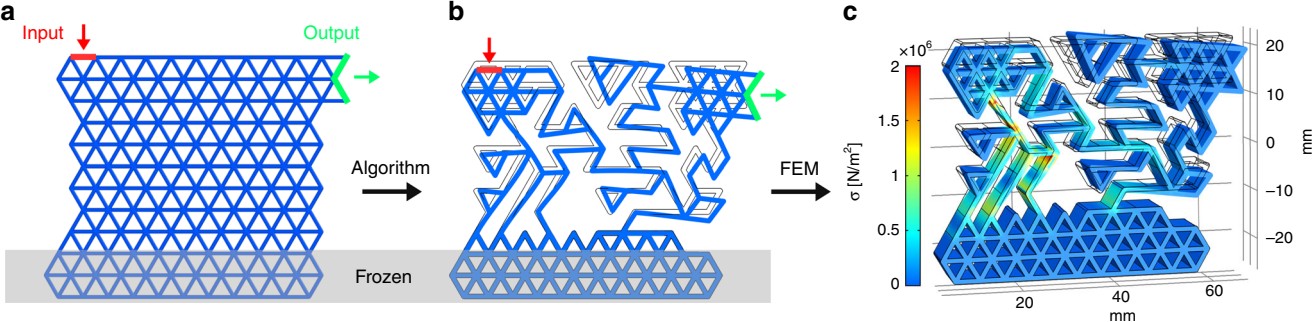

**Fig. 1 Schematic of automatic structure generation. a** The initial triangular lattice configuration **R**$_{IS}$. **b** The optimized structure obtained with DEM and its modeled response upon input displacement. **c** The same structure movement simulated by FEM. The colormap displays the Von Mises stress.

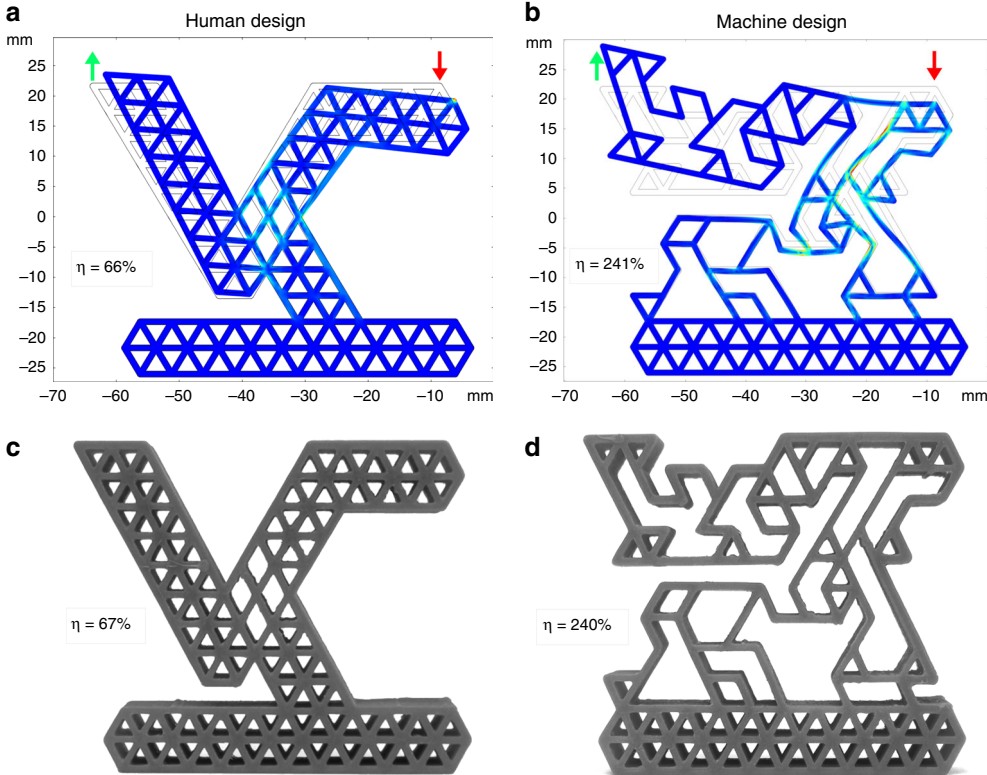

**Fig. 2 Automatic design achieves efficient anti-parallel movement.** Comparison of (**a**) human-designed and (**b**) machine-designed structures for FEM simulated and (**c**, **d**) the corresponding 3D printed realizations. The color represents the Von Mises stress with the same colorbar as in Fig. 1c. Red (green) arrows indicate the direction of the desired input **t**$_{inp}$ (output **t**$_{out}$). Resulting efficiencies are reported in each panel. For movement visualization see Supplementary Movie 2.

**Non-linear dependence of efficiency on input displacement**. The analysis presented in Fig. 2 and Supplementary Fig. 3 relied on an initial triangular mesh, but the algorithm can also be performed using a random mesh, as described in "Methods". The efficiency obtained using a periodic and a random lattice is compared in Fig. 4 for the same input movement. We observe that under a relatively small input displacement the random lattice based metamaterial outperforms the one obtained from a triangular lattice (Fig. 4a), but for a larger input displacement the two lattices yield similar efficiencies (Fig. 4b, d).

We investigate the origin of this observation by measuring the efficiency as a function of the input displacement. As shown in Fig. 4c, the efficiency is a non-linear function of the input displacement, saturating to a constant value at large enough displacements. In the random lattice case, the efficiency increases rapidly at very small displacement, while in the triangular case

such increase occurs at much higher displacements. The non-linearity results from a crossover between compression dominated to bending dominated behavior through local buckling instabilities which can produce large displacements at relatively low forces, acting as a multiplicative factor for the efficiency.

**Automatic design under force-based input/output**. The strategy employed to find the most effective structure under displacement-based input/output conditions can be extended to the case of a force-based input. To illustrate this, we consider, as a human-designed reference configuration, one of the metamaterial machines provided in ref. [36] reproducing the pliers, the most common traditional hand tool used to hold objects. The mechanical movement of the pliers consists of a pair of levers joined at a pivot point with short jaws on one side and longer

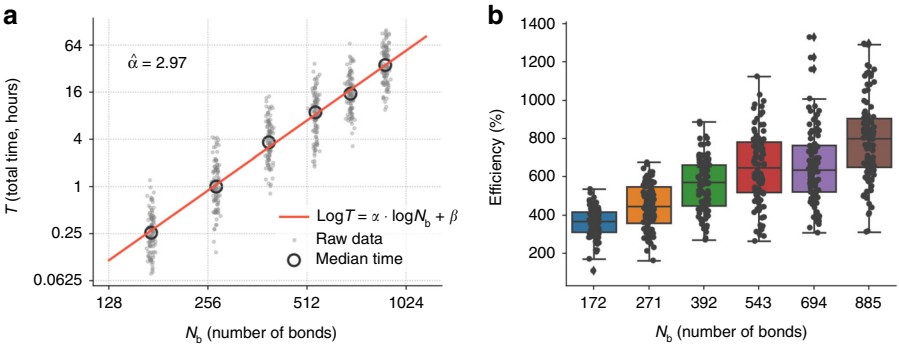

**Fig. 3 Scaling of the algorithm. a** Total execution time, in hours, versus number of bonds of a series of six increasingly large regular lattices. Empty large circles display the median execution time. Small gray dots show the raw data over 100 independent simulations. The red line is a linear fit in logarithmic space. The panel shows that the algorithm scales approximately as the cube of the number of bonds. **b** Boxplot of displacement-based efficiency as a function of number of bonds. The underlying raw data is also shown as black dots. Error bars in boxplots indicate minimum and maximum quartiles, boxes are first and third quartiles. The panel shows that efficiency initially increases with $N_b$ to then saturate, but there is great heterogeneity withing a fixed lattice size.

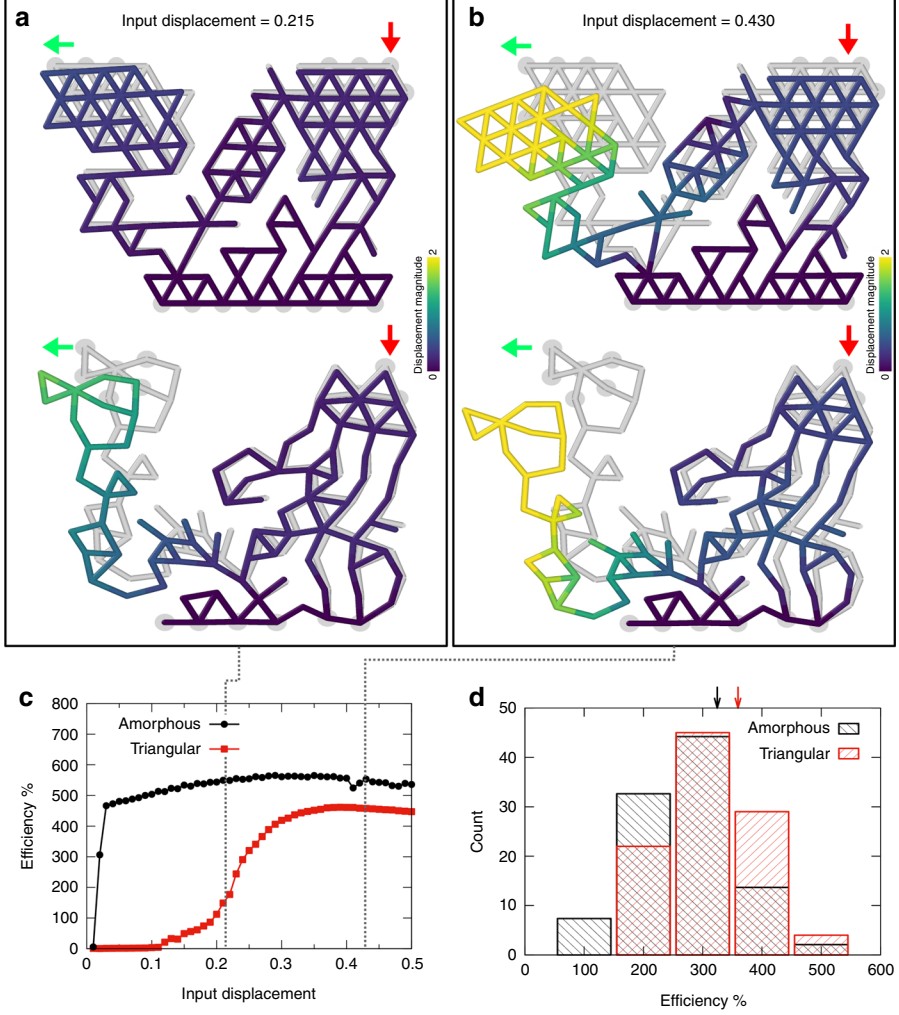

**Fig. 4 Lattice geometry and buckling effects. a** example of actuators with orthogonal input–output movement, obtained starting from a triangular (top) and amorphous (bottom) lattices. **b** The same as (**a**) but with a larger displacement of input nodes. Red (green) arrows indicate the direction of the desired input $\mathbf{t}_{inp}$ (output $\mathbf{t}_{out}$). **c** Efficiency as a function of the input displacement for the same two actuators in (**a**),(**b**). Error bars are standard deviations. **d** Efficiency distribution for 100 generated actuators using triangular or amorphous lattices. Arrows report the average efficiency for the two cases.

handles on the other one. The metamaterial machine proposed by Ion et al. consists of different unit cells which are softer near the fulcrum, to achieve more flexibility, and more rigid in the rest of the structure[36]. Pressing the handles (double anti-parallel input point application) results in the closure of the jaws to hold the object (double anti-parallel output) symmetrically along the $y$-axis passing through the fulcrum.

We tune our method to reach a target force-based efficiency, by applying constant input forces on two sets of handles nodes and measuring the force on a set of gauge springs, placed in contact with the gripping jaw nodes. The efficiency is then defined as in Eq. ((3)).

We implement DEM simulations on the human-designed pliers with suitable parameters that allow mechanical response within the linear elastic regime, choosing a relatively high gauge spring stiffness ($k_{ext} = 10.0$) and a relatively small input force ($F_{ext} = 0.01$). The goal is to achieve a large output force in response to a small input force. The force-based efficiency reached with these parameters is 22%. In order to visualize the real space displacements, we perform simulations also with a softer gauge spring (i.e. $k_{ext} = 0.01$, see Fig. 5a). Next, we run our method to automatically generate a pair of pliers with the same input parameters as the human-designed pliers ($k_{ext} = 10.0$ and $F_{ext} = 0.01$) but with improved efficiency in terms of force propagation. In practice, we request that the force-based

efficiency is more than double of the human-designed one ($\eta_f = 50\%$). Given the symmetry of the system along the line passing trough fulcrum, we simulate only the upper half of the system and subsequently mirror the structure in the bottom half. The starting structure is the rectangular fully connected network of beams with an area containing the human solution. During the minimization, the position of the pivot node is kept constant and the same is done for the $y$ coordinates of all the nodes in the symmetry line. The result is shown in Fig. 5b, where again we set $k_{ext} = 0.01$ to visualize the real space displacements.

We also investigate how the choice of $F_{ext}$ and $k_{ext}$ affects the resulting force-based efficiency. Fig. 5c reports the variation of the efficiency $\eta_f$ as a function of the output spring stiffness for fixed $F_{ext} = 0.01$ below and above our selected value of $k_{ext} = 10$ for both the human and machine-generated structures reported in panel a and b. For a given input force we observe a saturation for large output spring stiffness values, with similar trend for both the solutions. The estimate of the variation of the force-based efficiency as a function of input force for both the human and machine-generated pliers mechanism is reported in Fig. 5c. For input forces in the range $10^{-4}–10^{-2}$ the trend is constant for the human pliers and slightly increasing for the machine design, where we observe a decrease in the efficiency going above $F_{ext} = 10^{-2}$, due to buckling effects.

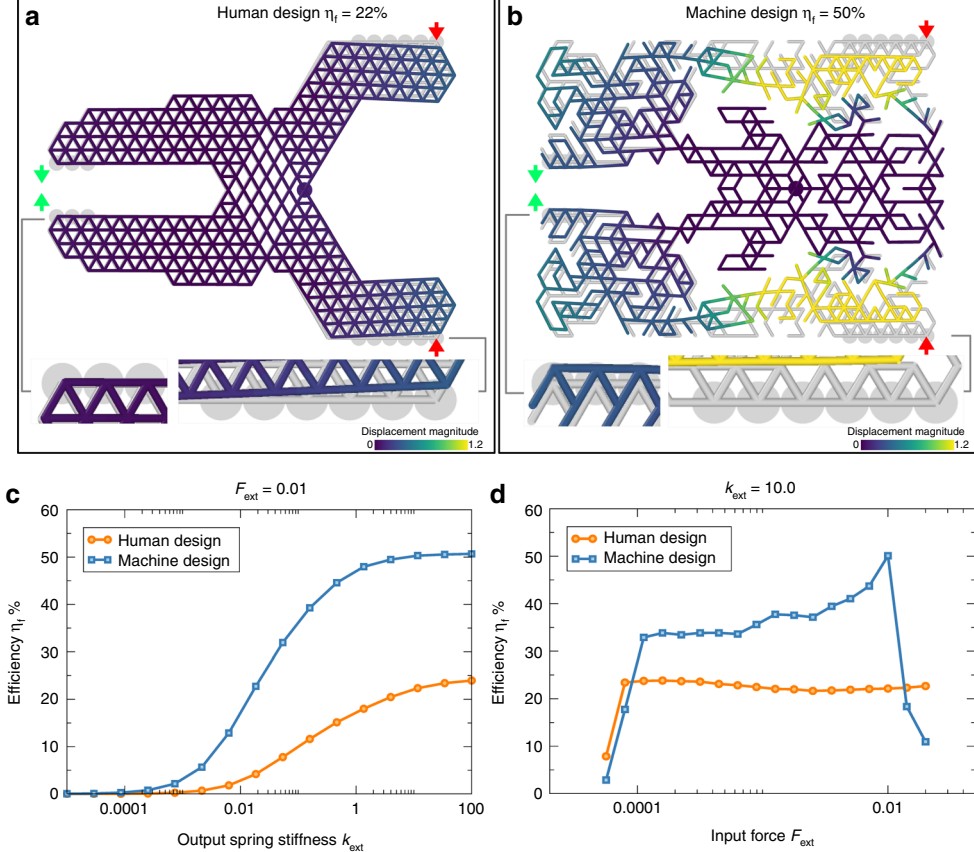

**Fig. 5 Machine-designed pliers achieve higher force-based efficiency than human-designed ones. a** Human-designed pliers and corresponding and (**b**) automatic-designed ones obtained with DEM simulations. The machine-designed configuration is compared with the human configuration for $F_{ext} = 0.01$ and $k_{ext} = 10.0$. To be able to visualize the displacements, we report both structures minimized with same input force $F_{ext} = 0.01$ but with lower output spring stiffness $k_{ext} = 0.01$. Elements are colored according to their displacement magnitude with respect to the rest configuration underlined in gray. The input (output) nodes are marked with bigger circles on the right (left) of the frozen pivot point (mark at the center). The blowups of the lower input and output nodes highlight the displacement field. Red (green) arrows indicate the direction of the desired input $\mathbf{t}_{inp}$ (output $\mathbf{t}_{out}$). **c** Variation of the force-based efficiency as a function of the output spring stiffness $k_{ext}$ for fixed value of $F_{ext} = 0.01$ and (**d**) as a function of the desired input force $F_{ext}$ for fixed value of output spring stiffness $k_{ext} = 10.0$.

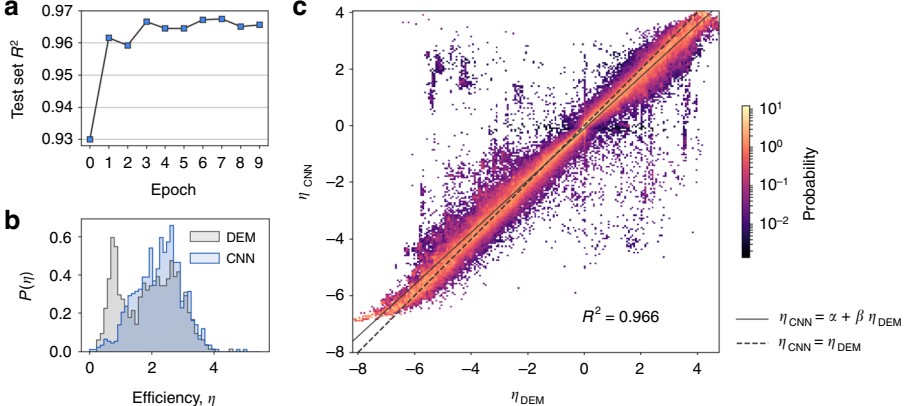

**Fig. 6 Machine learning can be used to predict efficient and design structures. a** Coefficient of determination $R^2$ of the test set during 10 training epochs. **b** Distribution of efficiency for configurations generated with the DEM model (gray) and the CNN model (blue). Notice that the efficiency of the resulting optimized configurations are always measured with DEM. **c** Density plot of the efficiency $\eta_{DEM}$, obtained through the DEM model, and $\eta_{CNN}$, obtained from the CNN model, showing that the CNN model is able to predict the DEM model efficiency values well. The solid line represents a linear fit, whose coefficient of determination is $R^2 = 0.966$. The dashed line is included as a guide to the eye only.

**Efficiency is predicted by machine learning**. The large set of metamaterial configurations we obtain offer a possibility for further insight when regarded as a dataset to be inquired. In this setting, one can naturally pose several questions related to how changes in structure relate to changes in efficiency. Here we assess whether static images of the configurations can be used to infer their efficiencies without the need of performing DEM or FEM simulations. To respond to this question, we train a Convolutional Neural Network (CNN) to perform image regression, that is, to predict the efficiency of a configuration from an image of its layout. Notice how this differs from the more standard use of CNN for image classification. Using a large number of configurations ($N \simeq 10^6$), we achieve an accuracy of $R^2 = 0.966$, see Fig. 6a, c and "Methods" for details.

**New structures can be generated through machine learning**. We take our ML-framework one step further and ask whether we can use our CNN model to generate new configurations from scratch by substituting the spring-mass model efficiency estimation step with predictions from the CNN model. In other words, we use the same Monte Carlo strategy described above but, at each step, instead of measuring the efficiency of the proposed configuration with DEM or FEM, we estimate it with the CNN model. Interestingly, the procedure is successful and we are able to generate new, efficient configurations with an approximately 100-fold speed up with respect to the DEM model. We notice, however, that we still need the DEM to train the CNN, hence a true speed up could be obtained only by suitable combinations of DEM and CNN. The CNN-generated configurations have a distribution of efficiency similar to that of the metropolis-generated ones, see Fig. 6b. To ensure that the obtained configurations are different from the ones used to train the CNN, we measure the minimal distance between each ML-generated configuration and all the configurations used during training, and find that the typical distance is of around 65 bonds.

**Functional regions can be identified with machine learning**. The CNN model is trained using spatial information (images of structures) and functional information (efficiency values). These two sources of information get effectively coupled in the trained CNN model and can be further exploited to identify functionally relevant regions. The gist of our method consists in feeding slightly perturbed images of a structure to the CNN, see Methods for details.

Fig. 7 shows an example of a configuration, highlighting the regions that according to the CNN would lead to significant changes in efficiency when adding (Fig. 7a) or removing (Fig. 7b) a bond. To confirm that these are indeed functionally relevant regions of the structure, we perform an analogous computation with the DEM model and show the results in Fig. 7c, d. Visual comparison with Fig. 7a, b shows that the CNN model can identify the key functional regions of the structure. We systematically quantify this effect over fifty independent runs, and find a high correlation between changes in efficiency predicted by the CNN and the DEM model (see Fig. 7i). In this example, the CNN model is tested with structures which are not part of the training data, but are nonetheless generated with the same underlying lattice, and are thus to some extent similar to the training data.

In the second example displayed in Fig. 7e–h, we go one step further and ask whether the same CNN model can identify functional regions when shown structures generated with larger sized lattices. In particular, the CNN model is trained with structures of a 172 bonds lattice, but it is later used to identify functional regions in structures generated with a 694 bonds lattice. Fig. 7e–h shows a structure on the 694-bond lattice: the CNN model correctly identifies the bottom left region as leading to lower efficiency when bonds are added to the structure. While, as expected, the correlation between the predicted and actual changes in efficiency is considerably lower than in the first example, there is a systematic bias towards positive values of $R$, which in some cases reaches values as high as 0.4 (Fig. 7j), which is remarkable for a prediction on a lattice that the CNN was not exposed to during training.

It is worth noting that the DEM calculations in Fig. 7c, d, g, h serve as ground truth to validate the method in this case, but they are also computationally more costly when switching from a 172 to a 694 bonds lattice, see Fig. 3. Being able to identify functionally relevant regions using a CNN model trained on smaller, simpler lattices opens the door to a myriad of possibilities, from lowering the rejection rate in the MC algorithm to combining human and machine efforts into the generation of hybrid structures.

## Discussion
In this paper, we have proposed algorithms for the automatic design of MMAs with a broad set of possible movements and efficiency—exceeding those of human-designed solutions. In its first implementation, the algorithm exploits a discrete elements

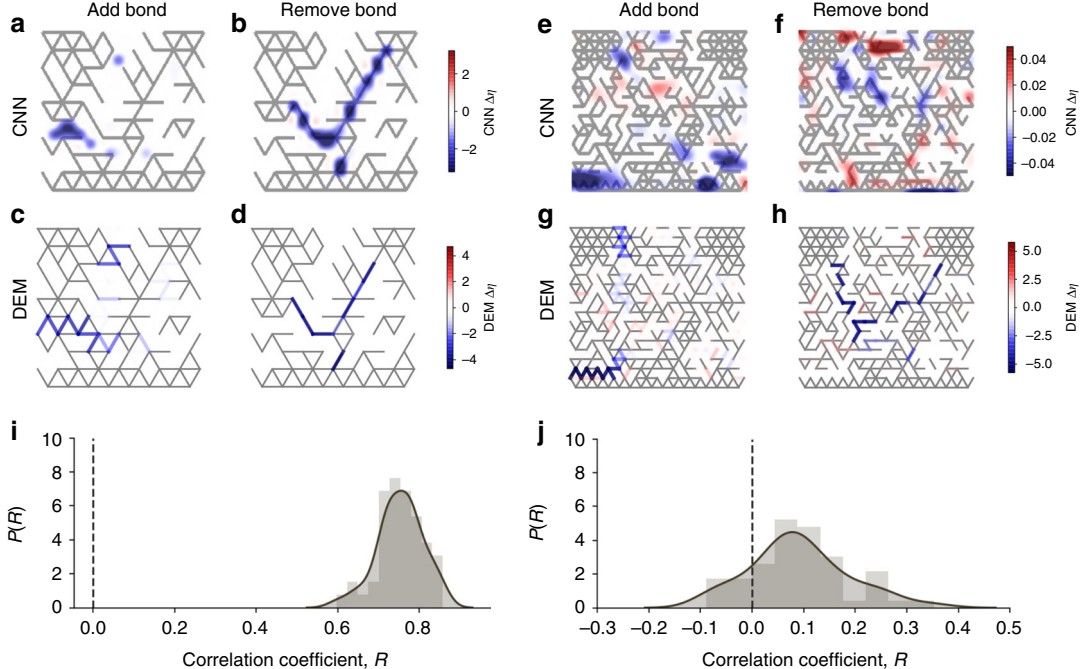

**Fig. 7 Functional heatmaps identify mechanically relevant regions of structures.** Heatmaps showing the spatial distribution of the changes in efficiency associated with bond addition (**a**, **c**) or bond removal (**b**, **d**), as computed by the CNN model (**a**, **b**) or the DEM model (**c**, **d**), in one example structure that lays on a 172-bond lattice. Blue shades correspond to regions that lead to lowered efficiency, while red regions lead to increased efficiency. **i** Distribution of the correlation coefficient $R$ between CNN and DEM values of $\Delta\eta$, see Methods for details **e–j** Same, for a structure of the 694-bond lattice, but still using the CNN model trained with 172-bond structures. The figure shows that information of a smaller lattice can be transferred to a larger one and still be used to identify functional regions of structures.

model to obtain an approximation of the mechanical efficiency of the structures which is then used to drive a Monte Carlo search over the possible structures. FEM calculations are then used to confirm the efficiency of the optimized structures that can then be realized by 3D printing. Here we have concentrated on two dimensional actuators but an extension to fully three dimensional models is also possible using the same strategy employed here.

It is interesting to compare our algorithm with previous approaches that optimized allosteric effects in random spring networks[37–39]. In contrast with these studies, we have concentrated in a system where angular bonds are included because this is fundamental to be able to reproduce mechanical features of real beams. Indeed the results of our discrete model compare well with FEM calculations. Absence of angular bonds can lead to floppy modes and loss of rigidity which are not present in real beam lattices. Beside this important point, previous approaches[37–39] also differ from ours in several technical aspects of the algorithm. The method proposed in Ref. [38] implements a Metropolis Monte Carlo algorithm at constant temperature with individual moves involving swapping of bonds so that the total coordination number is preserved. Our model does not have soft modes so we do not need to impose these restrictions which allows us to explore a wider phase space. Furthermore, we use simulated annealing which outperforms constant temperature Metropolis when the goal is to find a global minimum of a complex optimization problem. The algorithm presented in ref. [37] employs a sophisticated minimization strategy that is able to identify the bond whose removal would be most effective in terms of efficiency. When the efficiency landscape is complex; however, the algorithm might in principle be trapped into local minima, where no single bond removal would improve the efficiency.

Furthermore, we have employed deep neural networks to predict the efficiency of the actuator from its structure. Once

properly trained, the neural network can be used to create new structures without the need of performing DEM or FEM simulations. The use of machine learning to assist the automatic design of MMAs opens intriguing possibilities in terms of algorithmic speed, because it could potentially allow to design larger structures that can not be efficiently simulated by DEM. To this end, we explored the potential of deep neural networks in providing mechanical information on a given structure. Once trained, the network is able to identify from an image the regions of a structure where modifications would lead to an increase or a decrease in the efficiency. Identifying important functional regions from a disordered structure was also the object of recent investigations in the context of flow networks[43].

In conclusions, our work represents the first step toward the establishment of a reference library of elementary actuators (EA). More complex actuators can be subsequently obtained by the interlinking of multiple EA, with countless possibilities in terms of applications and flexibility. For instance, the algorithms could be useful to design moving parts in machines and robots, especially at small scales where the surface-to-volume ratio is very large, thus leading to dominant friction and wear. Benefits spans from the availability of ready-to-use EA, that will constitute a reference to engineers and material scientists, to the possibility of providing custom solutions for non-standard applications.

## Methods

**Triangular lattice based metamaterials**. We consider a triangular lattice configuration with coordinates $\mathbf{R}_{IS}$, which we can consider as the Inherent Structure[44]. Such configuration (see Fig. 1a) is mechanically stable and consists of $N_b$ beams of length $r = r_0$ connected to $N$ nodes. The position of the $i$-th node is $\mathbf{r}_i = (x_i, y_i)$, $\{\mathbf{r}_i\} = (\mathbf{r}_1, \mathbf{r}_2, \ldots, \mathbf{r}_N)$, and the distance between two nodes is $\mathbf{r}_{ij} = |\mathbf{r}_j - \mathbf{r}_i|$. We then select two far-apart (group of) nodes $i$ and $j$ which represent input and output regions, respectively, and we define two normalized vectors identifying their desired direction $\mathbf{t}_{inp}$ and $\mathbf{t}_{out}$.

**Random lattice based metamaterials**. Random metamaterial configurations are obtained starting from random configurations of disks brought at mechanically stable jamming point compressing the simulations box, as one standard protocol in jamming algorithms. Afterwards the center of each pair of overlapping disks is joined with a bond. As mentioned in ref. [37] this way of generating disordered structures ensures networks with mechanical properties which are stable and well understood.

**Efficiency**. The response of the metamaterial is monitored through its efficiency $\eta$,

$$\eta = \frac{\mathbf{t}_{\text{out}} \cdot (\mathbf{r}_j - \mathbf{r}_{0j})}{\mathbf{t}_{\text{inp}} \cdot (\mathbf{r}_i - \mathbf{r}_{0i})}, \tag{1}$$

where ($\mathbf{r}_{0i}$ and $\mathbf{r}_{0j}$) represent the original position of input and output nodes, respectively, and the dot products are averaged over the number of input (i) and output (j) nodes. One can envisage alternative definitions of efficiency, suitable to enforce the desired response in terms of a specific requirements of the optimized structure. We can provide two alternatives.

*Direction-based efficiency*: the search on metamaterial configurations is focused on the maximization of the output displacement toward the desired direction. To this purpose we generalize the dot product in Eq. ((1)) by a weight function $f(\gamma) = (2\cos(\gamma/2)^n - 1)$, $n \geq 2$, where $\gamma$ is the angle defined between the desired output direction $\mathbf{t}_{\text{out}}$ and the measured one. The resulting efficiency is

$$\eta_d = \frac{|\mathbf{r}_j - r_{0j}|f(\gamma)}{\mathbf{t}_{\text{inp}} \cdot (\mathbf{r}_i - \mathbf{r}_{0i})}, \tag{2}$$

that for $n \gg 2$ enforces the output motion along $\mathbf{t}_{\text{out}}$, while for $n = 2$ it is $f(\gamma) = \cos(\gamma)$, thus $\eta_d = \eta$.

*Force-based efficiency*: in this implementation, it is required that the exerted force on the input nodes is efficiently propagated to the output nodes towards the target direction. This is especially advisable when the actuator is expected to integrate with other mechanical parts, forming a larger mechanism. In this case, we apply a constant force on the input nodes, and we measure the force on the output nodes by means of gauge springs, acting as dynamometers along the $\mathbf{t}_{\text{out}}$ direction. This corresponds to adding the energy term $E_{\text{inp}} = F_{\text{ext}}[\mathbf{t}_{\text{inp}} \cdot (\mathbf{r}_i - \mathbf{r}_{0i})]$ to the input nodes, and $E_{\text{out}} = \frac{1}{2}k_{\text{ext}}(\mathbf{t}_{\text{out}} \cdot (\mathbf{r}_j - \mathbf{r}_{0j}))^2$ to the output ones during energy minimization, with $F_{\text{ext}}$ being the constant input force, and $k_{\text{ext}}$ being the spring constant of the output gauge springs. The force-based efficiency can then be straightforwardly defined as

$$\eta_f = \frac{\sum_j k_{\text{ext}} |\mathbf{r}_j - \mathbf{r}_{0j}| f(\gamma)}{\sum_i F_{\text{ext}}}, \tag{3}$$

where the sum at the nominator is over the output nodes and the sum at the denominator is over the input nodes.

**Optimization**. Once a suitable efficiency function is chosen, we maximize it by minimizing the cost function $\Delta = -\eta$. The minimization protocol makes use of the Monte Carlo (MC) method combined with an optimization algorithm: at each iteration step, from the present configuration with $\Delta = \Delta^0$, a trial configuration is obtained by removing or readding a randomly-selected beam. Input nodes and output nodes are discarded from pruning, as well as the set of selected nodes constrained against motion, i.e. frozen (see Fig. 1). We then displace the input nodes in the $\mathbf{t}_{\text{inp}}$ direction (or apply an external force to them in the case of $\eta_f$), perform a FIRE optimization[45], and by measuring the displacement of the output nodes (or the force on them through the monitoring springs) we evaluate the corresponding $\Delta^{\text{trial}}$. If $\Delta^{\text{trial}} < \Delta^0$ the removal/readding of the beam is accepted, otherwise it is accepted with a probability $P = \exp[-(\Delta^{\text{trial}} - \Delta^0)/T]$, where $T$ is a parameter acting as temperature in the MC dynamics.

At the beginning of each minimization, in order to explore the complex efficiency landscape[46], we perform 100 (accepted) MC steps of annealing with $T$ linearly decreasing from $T = 0.06$ (which is the threshold to consistently get $P \simeq 1$) to $T = 0$, and we subsequently let the algorithm evolve at the latter temperature. The whole procedure has been repeated in several runs using different random-number-generator seeds (see Supplementary Fig. 1).

It is worth to note that in a system with $N_b$ beams, the configuration space counts $\sim 2^{N_b}$ possible structures. In the example displayed in Fig. 1, $N_b = 203$ yields an extremely large number of configurations: $2^{203} \simeq 10^{61}$. Clearly, such an exponential scaling with $N_b$ severely limits the possibility of full exploration of the configuration space, and thus very fast methods to predict the efficiency of the trial structures are required. To maximize such exploration we have employed a combination of three different methods acting at different approximation levels, as described in the following.

**Discrete element model**. To obtain a fast and reliable estimation of the efficiency of a given structure we have made use of a simplified discrete element model (DEM) of the lattice, in which the total energy can be expressed as

$$E = \sum_i \sum_{j>i} \phi_2(\mathbf{r}_{ij}) + \sum_i \sum_{j>i} \sum_{k \neq j} \phi_3(\mathbf{r}_{ij}, \mathbf{r}_{ik}, \theta_{ijk}), \tag{4}$$

where the pairwise term is a spring potential with rest length $r_0$,

$$\phi_2(\mathbf{r}_{ij}) = k(\mathbf{r}_{ij} - \mathbf{r}_0)^2, \tag{5}$$

while the 3-body term introduces angular springs among the nearest-neighbor beams connected to the same node,

$$\phi_3(\mathbf{r}_{ij}, \mathbf{r}_{ik}, \theta_{ijk}) = \lambda \left[ \theta_{ijk} - \theta_{ijk}^0 \right]^2 \tag{6}$$

being $\theta_{ijk}$ the angle formed by beams $\overline{ij}$ and $\overline{ik}$, and $\theta_{ijk}^0$ the initial angle value in the triangular lattice. Both $\phi_2$ and $\phi_3$ act among first neighbors only, with 3-body neighbors dynamically recalculated at each step (see Supplementary Fig. 2). If not stated differently, we have employed the following unit-less parameters: $k = 5$, $\lambda = 0.1$, $r_0 = 1$.

The minimization of $E$ in the presence of frozen nodes (typically at the base of the structure) and of displaced input nodes, allows us to predict the response of the trial structure with a good compromise between speed and reliability. This will be therefore our reference method for the seek of structures with the highest efficiency (see Fig. 1b).

**Finite element method**. A realistic simulation of the mechanical response of a structure subject to an external force can be obtained by the finite element method (FEM). The method is based on a mesh representation of the object so that the continuum boundary value problem is transformed into a system of algebraic equations. The method is very accurate in the elastic regime, but works on the timescale of several seconds for our reference system (and rapidly increasing with the system size), and it is thus not suitable for the trial-error MC search of efficient structures. Rather, FEM has been employed before 3D printing as a final validation of the selected structures, generated through the more simplified and fast methods described above.

3D models of simulated structures have been produced by extrusion of each bond (see Fig. 1c). For FEM calculations we have employed COMSOL Multiphysics and COMSOL with MATLAB through its structural mechanics module[47]. All studies assume a linear elastic material with Young's modulus and Poisson's ratio estimated experimentally for bulk samples of interest. Results are obtained using Euler–Bernoulli beam elements and the in-built stationary studies calculation (a quasi-static solver).

**Machine learning**. *Deep learning model*: we take Resnet50 architecture as implemented in the python Keras library[48] and repurpose it to perform regression instead of classification by modifying the top layer with a single-unit dense layer.

*Data preparation*: we prepare $192 \times 128$ PNG images of structures obtained during one thousand DEM-based simulations with the desired $\mathbf{t}_{\text{out}}$ direction, including both accepted and rejected configurations. The CNN model needs to be able to recognize configurations that create a displacement in the desired direction, but also those that move in the opposite direction and whose efficiency is formally negative (to be able to avoid them during the Monte Carlo search). Since those configurations occur very rarely, we increase its sampling by running an additional one thousand DEM-based simulations aiming at the opposite $-\mathbf{t}_{\text{out}}$ direction. In total, we generate 1.163.733 images using the Python plotting library matplotlib.

*Model training*: we use a standard Adam optimizer and a mean squared error loss function. 75% of the runs are used as training data, while the remaining 25% are used for validation. We also implement a simple resampling strategy that renders the distribution of efficiencies approximately uniform over the training data. This mitigates the fact that, otherwise, the model predictions would be less accurate for very high-efficiency configurations, since these are less commonly found in our dataset. We train our modified Resnet50 CNN for 10 epochs in batches of size 32.

*Computational environment and model performance*: we use a 16-core computer equipped with a Tesla K20c GPU for the calculations shown in Fig. 6, and a Tesla V100-SXM2 for those shown in Figs. 3, 7. We measure the performance of the model using the $R^2$ value of linear regression between actual and predicted efficiency values.

**Human designed structures and 3D printing**. To provide a comparison of the automatically designed MMAs with conventional solutions, human-designed counterparts of the MMAs have been either created by us before running any simulation or taken from the literature, as to avoid any possible bias on the designer about the mechanisms leading to a high efficiency.

Samples are then produced by means of 3D printing, using the fused deposition modeling technique. In this method, the final structure is produced by laying down many successive thin layers of molten plastic. Each layer thickness is 0.8 $n_s$, where the nozzle size is $n_s = 0.4$mm The material used is NinjaFlex, a formulated thermoplastic polyurethane (TPU) material with super elastic properties.

Efficiency of 3D printed structures has been measured by manually moving the actuator input nodes by 5 mm with the aid of a caliper, while keeping fixed the bottom part holder. Images of the initial and the so displaced structure have then been captured by a camera, and displacements of input and output nodes have been measured by image processing. By applying Eq. ((2)) on the measured

displacements of input/output nodes, the efficiency $\eta_d$ was calculated and reported in Fig. 2 and Supplementary Fig. 1.

**Construction of functional heatmaps**. We construct functional heatmaps by feeding perturbed images of structures to the CNN model. A functional heatmap displays the predicted increase or decrease in the total efficiency when the structure is locally modified. For any given image of a structure, we construct two functional heatmaps: one that corresponds to the effects of adding bonds, and one that corresponds to the effects of removing bonds. However, because the lattice of the underlying training data and that of the structure under consideration might not coincide, we follow a lattice-agnostic procedure, as follows:

- Slide a perturbation square of $10 \times 10$ pixels over the image. The perturbation square is either all black (add) or all white (remove), depending on the heatmap we are constructing. We slide the square by 10 pixels each time, to avoid overlapping perturbations, but other more refined implementations are possible.
- In an image of $200 \times 200$ pixels, the $10 \times 10$ square can be placed in 400 different positions resulting in 400 perturbed images.
- Feed the 400 perturbed images to the CNN model, and predict the efficiency of the perturbed images.
- Subtract the predicted efficiency of the original image, obtaining the predicted change in efficiency of the 400 perturbed images.
- Construct a $20 \times 20$ array of the changes in efficiency values, recovering the position of the perturbation square.
- Zoom the image using cubic interpolation to obtain the final functional heatmap of size $200 \times 200$ pixels.

**Correlation between CNN and DEM functional heatmaps**. We use the correlation coefficient at the bond level as a simple measure of correlation between the efficiency changes predicted by the CNN model and those predicted by the DEM model, which act as ground truth. To infer efficiency change values at the bond level from a functional heatmap, we integrate the efficiency change values of the heatmap over the pixels that correspond to each bond. We define the correlation as the $R$-value of a linear least-square regression model as implemented by the *scipy. stats.linregress* python function.

## Data availability
Raw data used to generate the results of this papers are available at https://zenodo.org/record/3891249.

## Code availability
The codes used in this paper are available for the exclusive purpose of undertaking academic, governmental, or not-for-profit research. The main *Metamech* libraries can be accessed at https://github.com/complexitybiosystems/metamech. Example datasets and useful loader functions, to be used with the *Metamech* library are available at https://github.com/complexitybiosystems/metamech_datasets.

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

## Acknowledgements

This research has been supported by the European Research Council through the Proof of Concept grant 841640 METADESIGN. SZ also acknowledges support from the Alexander von Humboldt foundation through the Humboldt Research Award and thanks Friedrich-Alexander-Universität Erlangen-Nürnberg for hospitality. We gratefully thank F. Pezzotta for assistance with 3D printing.

## Author contributions

S.B., R.G., F.F.C. wrote the codes, performed simulations and analyzed data. D.R.K. performed FEM calculations. S.B.,R.G., F.F.C., S.Z. wrote the paper. S.Z. designed and coordinated the study.

## Competing Interests

The authors S.B., R.G., S.Z. declare the following competing interests: The University of Milan has filed a patent application related to the present work. Inventors: S. Bonfanti, R. Guerra and S. Zapperi. Patent status: pending. Date of application: 19/11/2019. Application number: 102019000021618. The patent concerns a design method for mechanical actuators. All other authors declare no competing interests.
