## [Peer Review File · Nature Communications]

Reviewers' comments:

Reviewer #1 (Remarks to the Author):

Automatic Design of Mechanical Metamaterial Actuators (Bonfanti et al.)

This manuscript proposes a machine-led approach to the design of mechanical metamaterials with actuation capabilities at desired target points. The authors' approach to the design of such metamaterial machines is novel and could lead to further research work in this area with high impact and importance. Therefore, I think this manuscript could be considered to be a good contribution to our current knowledge of additively-manufactured mechanical metamaterials; however, I believe that the presentation of the method, and the obtained computational and experimental results, should be improved before the acceptance of the manuscript for publication.

Main recommendations

- I believe that the authors need to present at least one more example (with extensive details) of the design generation process; such details should be included in the Supplementary Information to make sure that the work is replicable and reproducible for future investigators who could take this design generation approach to a next level.
- Within the area of mechanical metamaterials, this work can be categorized as metamaterial crystal lattices (of which a finite 3D shape is extracted). I believe that the literature review should mention some more recent publication, as it is important to provide readers with a background on the previous studies in this area. To cover the latest relevant literature, the following references are suggested, among other references that the authors may find to improve their literature review:
 - ♣ 'Nature Communications {2019}', volume 10, Article number: 5577 (2019) (Exploring multistability in prismatic metamaterials through local actuation).
 - ♣ 'Materials & Design {2019}', 183, 108128 (The least symmetric crystallographic derivative of the developable double corrugation surface).

Other recommendations

- In Figure 1, I suggest flipping the images (parts a, b, and c) horizontally in order for the output vectors to be pointing from left to right. This would make the sequence more natural and make it easier for the eye to follow.
- Figure 3 can be more informative, and graphically accurate (e.g., please try to make the centre of arches coincident with the intersection point (node) of line segments).

Reviewer #2 (Remarks to the Author):

The authors describe a computational technique for designing mechanical structures which convert a displacement or force actuation of a subset of "input" points to a desired displacement or force at specified "output" points. Structures are generated by pruning beams from a triangular lattice using a Monte Carlo algorithm that randomly removes or reinserts bonds to improve an efficiency criterion that measures the net conversion of input to output actuation calculated via a harmonic model (DEM) that reduces the structure to a combination of springs and angular springs. Algorithm-generated designs are shown to outperform a specific manually-generated design in full-blown finite element (FEM) tests of input-output conversion. The DEA calculations from a large set of MC runs are then used to train a CNN to predict the efficiency of a particular beam configuration, which enables a speedup of the Monte Carlo algorithm by avoiding additional DEA

calculations.

The analysis pursues some interesting ideas and the quantitative results presented are technically correct by my judgment. However, it is not clear to me that the authors have indeed achieved their stated claims in the abstract. The degree to which this work represents an advance over typical methods in the field is also questionable. I request the authors to respond to my major concerns below, perhaps with appropriate revisions, before I make a decision on the manuscript.

1. I am not convinced that training a CNN to replace DEM calculations "eliminat[es] the need for lengthy mechanical simulations". If I understood the procedure correctly, the training set was generated from thousands of MC runs for a particular input/output target combination which required the full DEM calculations. So, the possibility of generating fast additional designs (with a 100-fold speedup) only arises after a sizeable number of designs has already been generated. While it does seem that the CNN has "learned" patterns between beam configurations and efficiency, I do not see the utility of generating a large number of additional designs for a specific input/output combination once the demanding DEM computations have already generated many efficient designs. If the speedup had made accessible a substantially larger region of configuration space with much more efficient designs, perhaps the claim would be justified. However, Fig 6b shows that the designs generated by CNN are not substantially more efficient than the designs generated by DEM so I am not sure that the 100-fold speedup brought about by the deep learning algorithm is a consequential one. If the CNN-based MC simulations could generate efficient designs for arbitrary input/output combinations without having to be retrained, perhaps that would count as a consequential speedup.

2. The abstract also states that the high efficiency of the machine-generated designs is demonstrated by "3D printing of selected mechanical actuators". I did not see in the methods or the results section any description of actual measurements made on the the 3D printed structures. If the η values presented next to the images of 3D printed structures in Figs 4 and 5 are from experiments, details of the measurement must be provided.

3. The comparison to "human-designed structures" seemed somewhat subjective and not enough details were provided on how the designs were arrived at. For instance, it makes sense that the human design would be done before any machine-generated designs were seen, but did the human designer iteratively tweak their design by calculating efficiencies, or was it purely based on intuition? Also, the use of a triangular network of beams as the base design is natural for an algorithmic approach, but appears unnecessarily constraining for a human. It seems to me that a far better comparison would be between the machine designs from the proposed approach and designs generated by other algorithmic approaches. In particular, the MC procedure for generating desired networks is similar to allostery-inspired algorithmic design of mechanical actuators described in refs 10 and 11 of the manuscript; Ref 11 in particular also uses an MC procedure that adds and removes links, and the energy functionals used in both 10 and 11 are harmonic and similar to the DEM (they do not have angular bond stiffness but there is no fundamental obstacle to adding this). Do the authors have reason to believe that their approach outperforms past approaches? I understand that implementing other algorithms is likely beyond the scope of the work, but if the authors could attempt a qualitative comparison of the relative merits of their approach it would help bolster the conceptual advance of the manuscript.

Besides these major concerns, I have a few additional questions/comments which the authors may also choose to address:

- Do the authors have any estimates of the scaling of computational complexity of the proposed approach with system size? Does using the CNN change the complexity in a qualitative way?
- In the Methods, two efficiency choices are provided (direction-based and force-based) but these are not mentioned again. Are they both used later on?
- The authors mention that an advantage of the FEM method over DEM is that the force

propagation through the network can be generated in the former. However, forces on bond elements can also be generated from DEM by building the equilibrium or compatibility matrix [e.g. Pellegrino, *Int. J. Solids and Structures* **30**, 3025 (1993)].

- In the description of the training data set, it was not clear to me what aiming at "positive efficiency" or "negative efficiency" meant.
- In Fig 6b, are the CNN efficiencies the actual DEM efficiencies of the configurations generated by the CNN models, or are they the efficiency values predicted by the CNN?

Reviewer #3 (Remarks to the Author):

The paper "Automatic Design of Mechanical Metamaterial Actuators" introduces automated design methods for the allosteric mechanical response of triangular lattices of contacts via pruning and adding of bonds. Monte Carlo based methods are used to explore the space of available structures and a somewhat novel machine learning methods is used to quickly evaluate the mechanical response of the design. The paper is clearly written and accessible to a broad audience.

This paper joins a number of other studies in the area of optimization of network mechanics by pruning. My overall impression is a bit mixed. This work would have been ground breaking a few years back. However, recent advances in the field [1,2] go beyond mere optimization and derive a mathematical description of a network's allosteric response using persistent homology. While these two works are not yet published, they have been posted to the ArXiv and when they come out they will surely eclipse this contribution. While these works focus on flow networks, they follow the authors previous work on mechanical networks [3] (reference 10 in the text) and suggest that the same mechanism should describe allostery in mechanical networks. So in the context of the soon to be published literature, the current work falls a bit short in terms of the latest advances.

There are a couple of other issues. While the authors compare the efficiency of the results of their algorithm to a manual design (design effort time is not specified), it is already clear from the existing literature that optimization of allosteric mechanics can surpass human design. It would have been far more interesting if the authors compared their method to state of the art algorithms like the one introduced in [3].

An additional note is that the authors claim in the introduction (bottom of page 2) that their method deals with both ordered and disordered metamaterials. In the manuscript, however, I could only find results pertaining to triangular lattices.

To summarize, while the application of machine learning to evaluate the allosteric response of mechanical networks is somewhat novel, the MC methods in combination with Conjugant Gradient optimization do not seem to this reviewer to push the state of the art in the field, especially in light of recent advances in the understanding of the optimization of flow networks that will be published very soon.

(1) Jason W. Rocks, Andrea J. Liu, and Eleni Katifori, Revealing structure-function relationships in functional flow networks via persistent homology, arXiv:1901.00822, January 2019

(2) Jason W. Rocks, Andrea J. Liu, and Eleni Katifori, The hidden topological structure of flow network functionality, arXiv:1911.11606, November 2019.

(3) Jason W. Rocks, Nidhi Pashine, Irmgard Bischofberger, Carl P. Goodrich, Andrea J. Liu, and Sidney R. Nagel. Designing allostery-inspired response in mechanical networks. *Proceedings of the National Academy of Sciences*, 114(10):2520–2525, February 2017.

Response to reviewers' comments:

Reviewer #1 (Remarks to the Author):

This manuscript proposes a machine-led approach to the design of mechanical metamaterials with actuation capabilities at desired target points. The authors' approach to the design of such metamaterial machines is novel and could lead to further research work in this area with high impact and importance. Therefore, I think this manuscript could be considered to be a good contribution to our current knowledge of additively-manufactured mechanical metamaterials; however, I believe that the presentation of the method, and the obtained computational and experimental results, should be improved before the acceptance of the manuscript for publication.

We thank the reviewer for the positive remarks. We revised the manuscript improving its content including additional results to answer the constructive criticisms of the reviewer.

Main recommendations

• I believe that the authors need to present at least one more example (with extensive details) of the design generation process; such details should be included in the Supplementary Information to make sure that the work is replicable and reproducible for future investigators who could take this design generation approach to a next level.

We now include more examples as requested by the reviewer. In particular, we report now our reworking of metamaterial pliers (**see Fig. 5**), which were already proposed in the literature (see Ref. 36) as well as a new example in which the metamaterial actuator is constructed from a random lattice (**see Fig. 6**). We also provide more extensive details in the method section to ensure that our work is replicable. Furthermore, we provide access to the code (in GitHub) and to the raw data (in zenodo).

• Within the area of mechanical metamaterials, this work can be categorized as metamaterial crystal lattices (of which a finite 3D shape is extracted). I believe that the literature review should mention some more recent publication, as it is important to provide readers with a background on the previous studies in this area. To cover the latest relevant literature, the following references are suggested, among other references that the authors may find to improve their literature review:

- ♣ 'Nature Communications {2019}', volume 10, Article number: 5577 (2019) (Exploring multistability in prismatic metamaterials through local actuation).**
- ♣ 'Materials & Design {2019}', 183, 108128 (The least symmetric crystallographic derivative of the developable double corrugation surface).**

We thank the referee for mentioning these references that we now include in the paper. We improved the introduction by discussing in more details the most relevant recent publications on metamaterials (**see pages 2-3**). The reference list now includes additional relevant references, including those mentioned above (**Ref. 33 and 35**).

• In Figure 1, I suggest flipping the images (parts a, b, and c) horizontally in order for the output vectors to be pointing from left to right. This would make the sequence more natural and make it easier for the eye to follow.

We follow the advice of the referee and revise **Figure 1**.

• Figure 3 can be more informative, and graphically accurate (e.g., please try to make the centre of arches coincident with the intersection point (node) of line segments).

We revisit Figure 3 as suggested. Due to the additional results present in the revised manuscript, the figure has been moved into the supplement (it is now **Fig. S2**)

Reviewer #2 (Remarks to the Author):

The analysis pursues some interesting ideas and the quantitative results presented are technically correct by my judgment. However, it is not clear to me that the authors have indeed achieved their stated claims in the abstract. The degree to which this work represents an advance over typical methods in the field is also questionable. I request the authors to respond to my major concerns below, perhaps with appropriate revisions, before I make a decision on the manuscript.

We respond below to the remarks of the referee by performing additional work as suggested.

1. I am not convinced that training a CNN to replace DEM calculations "eliminat[es] the need for lengthy mechanical simulations". If I understood the procedure correctly, the training set was generated from thousands of MC runs for a particular input/output target combination which required the full DEM calculations. So, the possibility of generating fast additional designs (with a 100-fold speedup) only arises after a sizeable number of designs has already been generated. While it does seem that the CNN has "learned" patterns between beam configurations and efficiency, I do not see the utility of generating a large number of additional designs for a specific input/output combination once the demanding DEM computations have already generated many efficient designs. If the speedup had made accessible a substantially larger region of configuration space with much more efficient designs, perhaps the claim would be justified. However, Fig 6b shows that the designs generated by CNN are not substantially more efficient than the designs generated by DEM so I am not sure that the 100-fold speedup brought about by the deep learning algorithm is a consequential one. If the CNN-based MC simulations could generate efficient designs for arbitrary input/output combinations without having to be retrained, perhaps that would count as a consequential speedup.

We accept the criticism made by the referee. Indeed, it is not completely accurate to compare the performances of CNN and DEM. CNN is able to evaluate much faster than DEM the efficiency of a configuration, but to achieve this the network needs to be trained with DEM results. We thus reformulate the relevant discussion in the revised manuscript (**see page 17**).

The referee raises an important question: to which extents are the patterns learned by CNN transferable to other conditions that were not part of the training? In the previous version of the manuscript, we only showed that the CNN is able to estimate the efficiency of a new structure that was not part of the training set, but the conditions (input, output, lattice size) were the same as in the training. In the revised manuscript, we perform additional studies to illustrate the potential of CNN in providing information on a system with new conditions with respect to those employed for training. To this end, we train the CNN with configurations obtained with a small lattice size and then test it on larger size configurations. We visualize the results through heatmaps displaying CNN-predicted changes in the efficiency associated to the removal or addition of a bond. The CNN-generated heatmap can then be compared with the heatmap resulting from DEM calculations. The results show strong correlations between the patterns obtained by CNN and DEM for lattices of the same size. Interestingly, weaker but still significant correlations exist when the CNN evaluates larger-sized configurations. This result illustrates the potential transferring of learned patterns to novel conditions (**see Fig. 7 and the discussion at pages 18-19**).

2. The abstract also states that the high efficiency of the machine-generated designs is demonstrated by "3D printing of selected mechanical actuators". I did not see in the methods or the results section any description of actual measurements made on the the 3D printed structures. If

the η values presented next to the images of 3D printed structures in Figs 4 and 5 are from experiments, details of the measurement must be provided.

We now provide details of the measurements in the methods (see **page 8**).

3. The comparison to "human-designed structures" seemed somewhat subjective and not enough details were provided on how the designs were arrived at. For instance, it makes sense that the human design would be done before any machine-generated designs were seen, but did the human designer iteratively tweak their design by calculating efficiencies, or was it purely based on intuition?

The referee is correct in stating that the comparison with "human-designed structures" were subjective and possibly biased. Indeed for the provided examples, the design was based on intuition and we did the best we could do. To improve this comparison, we now consider an example of human design taken from the literature (in particular the pliers from Ion et al 2016). This should eliminate potential bias, since we were not the ones designing the actuator. We then use our method to redesign the same object and find again an improvement (see **Fig. 5** and the discussion at **pages 13-14**).

Also, the use of a triangular network of beams as the base design is natural for an algorithmic approach, but appears unnecessarily constraining for a human.

We thank the referee for this suggestion. In the revised version of the manuscript, we include a discussion of design based on random lattices (see **pages 12-13** and **Fig. 4**).

It seems to me that a far better comparison would be between the machine designs from the proposed approach and designs generated by other algorithmic approaches. In particular, the MC procedure for generating desired networks is similar to allostery-inspired algorithmic design of mechanical actuators described in refs 10 and 11 of the manuscript; Ref 11 in particular also uses an MC procedure that adds and removes links, and the energy functionals used in both 10 and 11 are harmonic and similar to the DEM (they do not have angular bond stiffness but there is no fundamental obstacle to adding this). Do the authors have reason to believe that their approach outperforms past approaches? I understand that implementing other algorithms is likely beyond the scope of the work, but if the authors could attempt a qualitative comparison of the relative merits of their approach it would help bolster the conceptual advance of the manuscript.

We have concentrated in a system where angular bonds are included because this is fundamental to be able to reproduce mechanical features of real beams. Indeed the results of our discrete model compare well with finite element calculations. Absence of angular bonds can lead to floppy modes and loss of rigidity which are not present in real beam lattices. Beside this important point, previous approaches (Ref 10 and 11 in the previous version of the paper) also differ from ours in several technical aspects that we now discuss in the paper. Ref 11 (now **Ref. 38**) implements a Metropolis MC algorithm at constant temperature. Individual moves in Ref. 11 involve swapping of bonds so that the total coordination number is preserved. Our model does not have soft modes so we do not need to implement this restrictive move allowing for a wider phase space. Furthermore, we use simulated annealing which outperforms constant temperature Metropolis. The algorithm presented in Ref. 10 (now **Ref. 37**) is a minimization algorithm that only consider sequential removal of bonds (no angular springs are considered there as well), so it is not designed to explore the full phase space and could converge to a local minimum (where all single bonds removal would not improve the efficiency). We discuss these points in the revised manuscript discussion section (see **page 20**).

Besides these major concerns, I have a few additional questions/comments which the authors may

also choose to address:

- Do the authors have any estimates of the scaling of computational complexity of the proposed approach with system size? Does using the CNN change the complexity in a qualitative way?

We now performed a size scaling analysis of the algorithm, which we report in the revised manuscript (see **Fig. 3** and the discussion at **page 11**). To assess the relevance of CNN in reducing computational complexity, we now explore transfer learning in which the CNN is trained with a small sized lattice and is then used to analyze larger scale lattices (see **Fig. 7** and the discussion at **page 18-19**).

- In the Methods, two efficiency choices are provided (direction-based and force-based) but these are not mentioned again. Are they both used later on?

We used both efficiency definitions. In the revised manuscript, we provide examples of structures obtained with both choices (see **Fig. 5** for a demonstration of force based efficiency).

- The authors mention that an advantage of the FEM method over DEM is that the force propagation through the network can be generated in the former. However, forces on bond elements can also be generated from DEM by building the equilibrium or compatibility matrix [e.g. Pellegrino, *Int. J. Solids and Structures* **30, 3025 (1993)].**

The referee is correct. We thus rephrased the sentence to avoid ambiguities (see **page 11**: "It is possible to obtain information about the stress propagation along the network from FEM or DEM simulations and thus identify the regions mostly involved in the mechanism actualization").

- In the description of the training data set, it was not clear to me what aiming at "positive efficiency" or "negative efficiency" meant.

This was instead unclear. We reformulate and expand the description in the revised manuscript (see **page 7**).

- In Fig 6b, are the CNN efficiencies the actual DEM efficiencies of the configurations generated by the CNN models, or are they the efficiency values predicted by the CNN?

We identified the configuration with CNN model but then recomputed the actual efficiencies with DEM. We specify this point in the caption of **Fig. 6**.

Reviewer #3 (Remarks to the Author):

The paper "Automatic Design of Mechanical Metamaterial Actuators" introduces automated design methods for the allosteric mechanical response of triangular lattices of contacts via pruning and adding of bonds. Monte Carlo based methods are used to explore the space of available structures and a somewhat novel machine learning methods is used to quickly evaluate the mechanical response of the design. The paper is clearly written and accessible to a broad audience.

This paper joins a number of other studies in the area of optimization of network mechanics by pruning. My overall impression is a bit mixed. This work would have been ground breaking a few years back. However, recent advances in the field [1,2] go beyond mere optimization and derive a mathematical description of a network's allosteric response using persistent homology. While these two works are not yet published, they have been posted to the ArXiv and when they come out they will surely eclipse this contribution. While these works focus on flow networks, they follow the authors previous work on mechanical networks [3] (reference 10 in the text) and suggest that the

same mechanism should describe allostery in mechanical networks. So in the context of the soon to be published literature, the current work falls a bit short in terms of the latest advances.

We do not respond to the issue of how groundbreaking is our work, since it would be a subjective statement. We acknowledge that the unpublished work discussed by the referee is very interesting. We are happy to cite it, but we feel that it goes into a different direction with respect to our work. While we do not doubt the relevance of scalar models to understand some aspects of the mechanical properties of materials (one of us has studied random fuse models for years), we also know that scalar models are inadequate to design mechanical actuators. This should be clear when we note that in a scalar model it is not possible to define concepts such as “up/down” or “left/right”. Moving into spring networks has also issues, due to the presence of floppy modes and the absence of bending rigidity for the beams. For this reason, these networks do not display buckling instabilities which are very important for mechanical metamaterials (see **Fig. 4c** for an illustration of the role of buckling for the efficiency). Working with a realistic elastic model was important in the context of our paper because our primary motivation was to introduce design methodologies that would allow us to optimize a structure that could then be 3D printed. In summary, results obtained with scalar or central force models are not always relevant for real mechanical metamaterials and generalization to tensorial elasticity was not straightforward.

There are a couple of other issues. While the authors compare the efficiency of the results of their algorithm to a manual design (design effort time is not specified), it is already clear from the existing literature that optimization of allosteric mechanics can surpass human design. It would have been far more interesting if the authors compared their method to state of the art algorithms like the one introduced in [3].

We have improved our comparison with human design, by comparing our results with a metamaterial actuator designed by other authors. Furthermore, as also discussed in the response to the first reviewer, we include in the revised manuscript a discussion comparing our method with other methods reported in the literature (see **page 20**). The algorithm presented in [3] is a minimization algorithm that only considers sequential removal of bonds (no angular springs are considered and no bond re-insertion). Hence, the method cannot be applied to our case. Furthermore, removing one bond at a time does not guarantee that the global minimum is always reached. The algorithm might in principle be trapped into a configuration where each single bond removal does not improve the minimum and yet another lower lying minimum could exist.

An additional note is that the authors claim in the introduction (bottom of page 2) that their method deals with both ordered and disordered metamaterials. In the manuscript, however, I could only find results pertaining to triangular lattices.

We agree with the reviewer and to address her/his concern we now show results obtained with random lattices (see Fig. 4 and the discussion at **pages 12-13**).

To summarize, while the application of machine learning to evaluate the allosteric response of mechanical networks is somewhat novel, the MC methods in combination with Conjugate Gradient optimization do not seem to this reviewer to push the state of the art in the field, especially in light of recent advances in the understanding of the optimization of flow networks that will be published very soon.

As we discussed above, the generalization of results for flow networks (or even spring network) to real mechanical networks is far from being trivial. Thus, while we appreciate the results mentioned by the reviewer, we would like to stress again that our work expands current knowledge along a different direction.

REVIEWERS' COMMENTS:

Reviewer #1 (Remarks to the Author):

My concerns associated with the previous version of this paper have been well-addressed by the authors; therefore, I would like to recommend this manuscript for publication.

Reviewer #2 (Remarks to the Author):

Having read the revised version of the paper and the responses to previous queries, I now have a clearer sense of the advances of the paper compared to previous approaches. To summarize:

- The authors include angular bond stiffness in their elastic energy functional unlike previous works which were restricted to harmonic springs between vertex pairs. While I don't see a fundamental obstruction to including angular constraints in the methods of the previous works, the present work appears to be the first to verify the applicability of optimization including angular bonds to experiments on cellular beam-based metamaterials for which these constraints are certainly relevant.

- The current Monte Carlo optimization scheme uses a larger range of moves compared to previous works (primarily, both adding and removing bonds are allowed; furthermore, simulated annealing is used in contrast to constant-temperature methods. Ideally, the work would be able to demonstrate the superiority of the approach over the previous ones in a quantitative manner, but I am satisfied with the comparison in the discussion section (but see my suggestions below for an improvement).

- The use of machine-learning methods does increase the accessible design space, but does not seem to significantly improve the design process in terms of efficiency achieved or computational requirements. However, the revision is open about these limitations, and has some suggestive results with regards to applying a CNN to computations on system sizes beyond those used in the training set. I think it is useful to have these results, including their limitations, reported for the benefit of other studies in a field that is likely to be a hotbed of research.

The revision significantly broadens the scope of the work, by including a non-self-designed motif; a comparison between triangular and random initial lattices; size-scaling of the computational complexity; and functional heatmaps. It also provides details of the experimental part which had been lacking, providing a comprehensive analysis from abstract design to real-world realization. With these improvements, I judge it to be a significant contribution which will advance the specific field, and also has potential insights to offer to related fields. I therefore recommend publication.

I did find a few minor issues which the authors might like to address before publication:

- In the "Construction of functional heatmaps" section of Methods, it would be helpful to either have a one-sentence definition of what a functional heatmap is, or refer to the forthcoming section ("Functional regions of a structure can be identified with machine learning") where the concept of functional regions is explained. Otherwise the reader is left in the dark until later.

- The statement "Furthermore, we use simulated annealing which usually outperforms constant temperature Metropolis." needs to be made more precise. It's not clear what "usually outperforms" means. A little more explanation and/or references to works which demonstrated the superiority of the former method over the latter (presumably applied to a different optimization problem) would make the statement meaningful.

- "loosing" -> "losing" (page 2)

REVIEWERS' COMMENTS:

Reviewer #1 (Remarks to the Author):

My concerns associated with the previous version of this paper have been well-addressed by the authors; therefore, I would like to recommend this manuscript for publication.

We thank the Reviewer for her/his positive evaluation.

Reviewer #2 (Remarks to the Author):

Having read the revised version of the paper and the responses to previous queries, I now have a clearer sense of the advances of the paper compared to previous approaches. To summarize:

- The authors include angular bond stiffness in their elastic energy functional unlike previous works which were restricted to harmonic springs between vertex pairs. While I don't see a fundamental obstruction to including angular constraints in the methods of the previous works, the present work appears to be the first to verify the applicability of optimization including angular bonds to experiments on cellular beam-based metamaterials for which these constraints are certainly relevant.

- The current Monte Carlo optimization scheme uses a larger range of moves compared to previous works (primarily, both adding and removing bonds are allowed; furthermore, simulated annealing is used in contrast to constant-temperature methods. Ideally, the work would be able to demonstrate the superiority of the approach over the previous ones in a quantitative manner, but I am satisfied with the comparison in the discussion section (but see my suggestions below for an improvement).

- The use of machine-learning methods does increase the accessible design space, but does not seem to significantly improve the design process in terms of efficiency achieved or computational requirements. However, the revision is open about these limitations, and has some suggestive results with regards to applying a CNN to computations on system sizes beyond those used in the training set. I think it is useful to have these results, including their limitations, reported for the benefit of other studies in a field that is likely to be a hotbed of research.

The revision significantly broadens the scope of the work, by including a non-self-designed motif; a comparison between triangular and random initial lattices; size-scaling of the computational complexity; and functional heatmaps. It also provides details of the experimental part which had been lacking, providing a comprehensive analysis from abstract design to real-world realization. With these improvements, I judge it to be a significant contribution which will advance the specific field, and also has potential insights to offer to related fields. I therefore recommend publication.

We thank the Reviewer for her/his recommendation.

I did find a few minor issues which the authors might like to address before publication:

- In the "Construction of functional heatmaps" section of Methods, it would be helpful to either have a one-sentence definition of what a functional heatmap is, or refer to the forthcoming section ("Functional regions of a structure can be identified with machine learning") where the concept of functional regions is explained. Otherwise the reader is left in the dark until later.

We add one sentence as suggested.

- *The statement "Furthermore, we use simulated annealing which usually outperforms constant temperature Metropolis." needs to be made more precise. It's not clear what "usually outperforms" means. A little more explanation and/or references to works which demonstrated the superiority of the former method over the latter (presumably applied to a different optimization problem) would make the statement meaningful.*

We expand the sentence. Constant temperature Metropolis is just not designed to find the global minimum of a complex optimization problem.

- *"loosing" -> "losing" (page 2)*

Corrected, many thanks.